

# Lymphoid-specific helicase inhibits cervical cancer cells ferroptosis by promoting Nrf2 expression

Weiwei Tie and Fenfen Ge

Department of Gynecology, Ningbo Medical Center Lihuili Hospital, Ningbo University, Ningbo, Zhejiang, China

## ABSTRACT

**Background**. Cervical cancer is a major cause of morbidity and mortality in women worldwide. The underlying mechanisms of its progression are not well understood. In this study, we investigated the role of lymphoid-specific helicase (HELLS) in cervical cancer.

**Methods**. We measured HELLS expression in cervical cancer and assessed its function using gain- and loss-of-function experiments. Cell viability was measured using the Cell Counting Kit-8 (CCK8) assay, and cell proliferation was analyzed using colony formation and EdU assays.

**Results**. We found that HELLS was significantly increased in cervical cancer and that its overexpression promoted cell viability ($P < 0.01$) and colony formation ($P < 0.001$). In contrast, si-HELLS suppressed these effects. Moreover, HELLS overexpression inhibited cell death induced by the ferroptosis inducer erastin ($P < 0.01$). Mechanistically, we found that HELLS promoted cervical cancer proliferation by regulating nuclear factor erythroid 2-related factor 2 (Nrf2)-mediated ferroptosis.

**Conclusion**. Our data suggest that HELLS promotes cervical cancer proliferation by inhibiting Nrf2 expression. Therefore, HELLS knockdown may be an effective treatment for cervical cancer.

## INTRODUCTION

Cervical carcinoma is one of the most prevalent tumor-related causes of death in women in the world (*Johnson et al., 2019*; *Vu et al., 2018*). The major cause of cervical carcinoma is well-known to be infection with the human papillomavirus (HPV) (*Burd, 2003*; *Olusola et al., 2019*). The risk of HPV infection persists throughout life. Approximately 20% of HPV-infected women develop pre-treatment lesions of the cervical intraepithelial neoplasia (CIN) type. Most women with persistent lesions will later clear the virus, which will lead to the regression of persistent lesions. If the virus is not cleared, persistent infection with high-risk HPV oncogenic genotypes (HPV 16, HPV 18, HPV 31, HPV 33, HPV 35, HPV 45, HPV 58) will lead to the development of CIN 3 lesions, which will lead to invasive cancer after a variable period of 8–13 years. One of the most important factors in cervical carcinogenesis is the persistent infection of HPV oncogenes (for more than 6–12 months)

Corresponding author
Weiwei Tie, tieweiwei2006@163.com

(*Sharma et al., 2023*; *Tit et al., 2017*). Although HPV vaccines and early screening can reduce the mortality of HPV-related cervical carcinoma (*Hancock, Hellner & Dorrell, 2018*; *Wang et al., 2020*), patients with cervical carcinoma, especially those with regional or distant metastasis, still face a poor prognosis due to limited therapeutic options. Despite the development of numerous drugs and natural compounds aimed at treating cervical carcinoma in recent years (*Sharma et al., 2023*), their effectiveness in clinical application continues to fall short of expectations. Therefore, it is essential to investigate new therapeutic strategies and potential target for cervical carcinoma, which demands a greater understanding of the pathophysiology and biological mechanism of cervical carcinoma progression.

Ferroptosis is a newly discovered type of programmed cell death characterized by an iron-dependent accumulation of lipid peroxidation (*Jiang, Stockwell & Conrad, 2021*; *Li et al., 2020a*). In recent years, a growing body of research has established a strong correlation between ferroptosis and tumor development (*Bano et al., 2022*). Specifically, three pivotal negative regulators of ferroptosis, nuclear factor erythroid 2-related factor 2 (Nrf2, encoded by nuclear factor erythroid 2 like 2 (NFE2L2)), glutathione peroxidase 4 (GPX4), and solute carrier family 7 member 11 (SLC7A11), emerge as critical factors in cancer development through their regulation of oxidative stress and lipid peroxidation. *Roh et al. (2017)* reported that cisplatin-resistant head and neck cancer cells are sensitive to ferroptosis induced by artesunate when Nrf2 is inhibited. *Wang et al. (2021b)* reported that high levels of SOX2 promote lung cancer cell ferroptosis by positively regulating SLC7A11 expression. *Wang et al. (2022)* demonstrated that Wnt/beta-catenin signaling alleviates gastric cancer cellular ferroptosis by targeting GPX4. Nrf2, a stress-responsive transcription factor, is increasingly implicated in mediating ferroptosis through its modulation of cellular antioxidant defenses, iron homeostasis, and intermediate metabolism. Furthermore, Nrf2 is well-established as a regulator of two key targets, SLC7A11 and GPX4, both crucial for initiating the process of ferroptosis. *Tossetta & Marzioni (2023)* demonstrated that the Nrf2/KEAP1 pathway has a significant impact on cervical and endometrial cancers. It influences the expression of its downstream genes. Furthermore, this pathway is instrumental in the development of resistance to chemotherapy treatments of cervical and endometrial cancers (*Tossetta & Marzioni, 2023*). In another review, they also describe the important role of Nrf2 in the development of prostate cancer (*Tossetta et al., 2023*). However, the function of Nrf2-mediated ferroptosis in the development of cervical carcinoma is still unclear.

Lymphoid-specific helicase (HELLS), belonging to the SNF2 family of chromatin remodeling enzymes, has previously demonstrated elevated expression in multiple cancer types, including colorectal cancer, hepatocellular carcinoma (HCC), and lung cancer. *Liu et al. (2019)* reported a significantly upregulation of HELLS in colorectal cancer (CRC) and identified correlations with clinicopathological parameters. High HELLS expression served as an indicator of an adverse prognosis for patients with CRC. Another study demonstrated that a significant overexpression of HELLS in HCC, and HELLS was identified as a key mediator of epigenetic silencing for numerous tumor suppressor genes in HCC, encompassing E-cadherin, FBP1, IGFBP3, XAF1, and CREB3L3 (*Law et al., 2019*).

*Zhu et al. (2020)* reported that HELLS and ICAM1 could potentially serve as pivotal genes associated with the tumor progression of lung cancer. However, the function of HELLS in the development of cervical carcinoma is still unclear. In the current study, we clarify the function of HELLS in cervical cancer progression. Elevated HELLS expression was observed in cervical cancer, and its overexpression was associated with increased cell viability and colony formation while inhibiting cell death induced by ferroptosis. Mechanistically, HELLS was found to promote cervical cancer proliferation through the regulation of Nrf2-mediated ferroptosis.

## MATERIALS AND METHODS

Portions of this text were previously published as part of a preprint (https://www.researchsquare.com/article/rs-2689719/v1).

### Patients sample

A total of 15 pairs of cervical cancer tumor tissues and paired normal tissues were obtained by surgical resection. Patients with other cancers and received preoperative chemo/radiotherapy were excluded. This study was approved by the Research Ethic Committee of the Ningbo Medical Centre Lihuili Hospital (DYLL2018079, 2018). All patients signed informed consent prior to participating in the study.

### Immunohistochemistry (IHC)

Immunohistochemistry staining for HELLS was carried out. A total of 4-$\mu$m sections of cancer tissues were rehydrated and incubated with anti-HELLS (1:100, ab3851, Abcam, Cambridge, UK) overnight. The next day, tissue sections were washed with PBS, incubated with goat-anti mouse HRP-conjugated (1:1000, ab6721, Abcam, Cambridge, UK) for 1 h at 37 °C, and visualized by diaminobenzidine (DAB). The sections were counterstained byhematoxylin (Solarbio, Beijing, China).

### Cell culture andtreatment

ICE cells and cervical cancer cell lines C4-1, HeLa, Caski, and SiHa were purchased from Chinese Academy of Science (Shanghai, China). All cells were grown in RPMI-1640 medium (Gibco, Waltham, MA, USA) supplemented with 10%FBS. All cell lines were cultured in an incubator with 5% $CO_2$ at 37 °C.

To overexpress HELLS, The HELLS lentiviral plasmid was purchased from Genechem (Shanghai, China), and was transfected into C4-1 cells using Lipofectamine™ 3000. The cells were selected by 3 $\mu$g/mL puromycin for one month after 48 hours' transfection.

To silence HELLS or Nrf2, si-HELLS (siRNA ID: 145350) and si-Nrf2 (siRNA ID: 107969) were purchased from Thermo Fisher (Waltham, MA, USA), and SiHa or C4-1cells were transfected by Lipofectamine® RNAiMAX Transfection Reagent (Invitrogen, Carlsbad, CA, USA), in accordance with the manufacturer guidelines.

To inhibit apoptosis or ferroptosis, SiHa cell with si-HELLS were stimulated with Z-VAD-FMK (10 $\mu$M) or ferrostatin-1 (0.5 $\mu$M). To induce ferroptosis in cervical cancer cells, HELLS-OEC4-1cells were treated with erastin (20 $\mu$M) for 24 h.

## Quantitative real-time PCR

Total RNA was extracted from cervical cancer tissues and cells using Trizol reagent (Invitrogen, Carlsbad, CA, USA). The SMART PCR cDNA Synthesis Kits were used for reverse transcriptional PCR (Clontech, Mountain View, CA, USA). Quantitative real-time PCR (qRT-PCR) was carried out with SYBR Green incorporation (Thermo Fisher, Waltham, MA, USA). The $2^{-\Delta\Delta Ct}$ method was utilized to determine the fold changes of RNA transcripts, and the actin gene was employed as a reference gene.

## Cell counting kit-8

The cell viability of cervical cancer cells was analyzed by using the Cell-counting kit-8 (CCK8) (Thermo Fisher, Waltham, MA, USA). Cervical cancer cells ($2 \times 10^4$ cells/ml) were seeded on 96-well plate (100 µl per well), and cells were cultivated for five different times under diverse treatments. The CCK-8 test was carried out at each interval of time in accordance with the manufacturer's instructions.

## 5-ethynyl-2′-deoxyuridine (Edu)assay

EdU assay was performed to measure the DNA synthesis of cervical cancer cells. Cells were seeded at $2 \times 10^4$ cells/ml in 96-well plates (100 µl/well) and fixed with 4% paraformaldehyde for 10 min. Then, cells were permeabilized with 0.5% Triton-X-100 in PBS for 20 min and blocked with 1% BSA in PBS for 30 min at room temperature. Next, cells were treated with 10 µM EdU for 1 h at 37 °C with 5% $CO_2$. After that, the Click-iT® reaction cocktail (Thermo Fisher, Waltham, MA, USA) was added and cells were incubated for 30 min at room temperature. Finally, cells were stained with Hoechest for 5 min and imaged using a Multiskan FC microplate reader (Thermo Fisher, Waltham, MA, USA).

## Western-blotting analysis

Western-blotting was performed as our previous study (*Tie & Ge, 2021*), and the antibodies used were as follows: anti-HELLS (1:1000, ab3851, Abcam, Cambridge, UK), anti-Nrf2 (1:1000, ab137550, Abcam, Cambridge, UK), and anti-actin (1:5000, ab8224, Abcam, Cambridge, UK) overnight at 4 °C.

## Colony formation assay

Colony formation was performed as our previous study (*Tie & Ge, 2021*), Briefly, cervical cancer cells were seeded in 10 cm-plates at a concentration of 1,000 cells/well and cultured normally for 14 days in RPMI 1,640 with 10% FBS. The colonies were fixed and stained with 0.4% crystal violet (Sigma, Burlington, MA, USA) and methanol before being counted.

## Statistical analysis

In the current study, statistical analyses were carried out by SPSS 22.0 software (SPSS, Chicago, IL, USA) (*Duricki, Soleman & Moon, 2016*). The mean ± standard deviation is used to represent all the data, the differences between two groups were analyzed by Student's $t$-test, and a one-way ANOVA was used to investigate differences between more than two groups. $P$ values < 0.05 were considered significant.

## RESULTS

### HELLS was upregulated in cervical carcinoma

To explore the genes closely related to the occurrence of cervical cancer, four cervical carcinoma high-throughput sequencing datasets (GSE63678, GSE122697, GSE63514, and GSE192897) were obtained from the Gene Expression Omnibus (GEO)database, and the common different expression genes were obtained through Venn overlap analysis. Figure 1A showed that seventeen common genes (HELLS, MELK, EZH2, CENPF, KIF2C, CCNB1, TYMS, MAPK10, CEP55, BIRC5, STAT1, CDC20, ESR1, CXCL8, CXCL9, CXCL11, and CXCL1) were obtained. Among these genes, HELLS is the one with the most significant differences. To confirm the function of HELLS, the expression of HELLS in pan-cancer was assessed by the Tumor Immune Estimation Resource (TIMER) database (http://timer.cistrome.org/) (Li et al., 2020b). Results showed that HELLS was significantly upregulated in most cancers, including cervical carcinoma (Fig. S1). Then, the HELLS expression was further assessed in the TCGA database. Figure 1B showed that HELLS was significantly upregulated in the cervical carcinoma group ($N = 306$) compared with the control group ($N = 13$). Next, HELLS expression was assessed in fifteen cervical carcinomas and paired normal tissues. In line with the results of bioinformatics analysis, we also found that HELLS was significantly up-regulated in the cervical carcinoma group compared with the normal control (Fig. 1C). Then survival analysis was performed on HELLS in Cervical squamous cell carcinoma (CESC) by GEIPA2 software, which revealed high HELLS expression was unfavorable to patient prognosis, regarding OS (overall survival), DFS (disease free survival) (Figs. 1C and 1D). Moreover, IHC staining also showed that HELLS was significantly up-regulated in the cervical carcinoma group compared with the normal control (Fig. 1E).

### HELLS facilitates cervical cancer cell proliferation

To verify the function of HELLS in cervical cancer, the expression of HELLS was analyzed in four cervical cancer cell lines (Caski, SiHa, HeLa, c4-1). Figures 2A and 2B showed that HELLS was significantly increased in all cervical cancer cell lines compared with the normal ICE cells. Among these cell lines, SiHa cells showed the highest HELLS level, whereas C4-1 cells showed a relatively lower HELLS level. Therefore, to investigate the function of HELLS in cervical cancer *in vitro*, we overexpress HELLS (HELLS-OE) in C4-1 cells by lentivirus. Figures2C and D showed that HELLS expression was significantly increased by lentivirus. Next, the cell viability was assessed by CCK8. Figure 2E showed that C4-1 cell viability was significantly increased in the HELLS-OE group. Next, cell proliferation was assessed by the colony formation assay. Similarly, colony formation data showed that the proliferation abilities of C4-1 cells were significantly increased in the HELLS-OE group (Fig. 2F). Moreover, the cell proliferation was analyzed by EdU assay. Figure 2G showed that HELLS markedly promote C4-1 cells proliferation.

Subsequently, HELLS was knocked down in SiHa cells by siRNA interference. Figures 3A and 3B showed that HELLS expression was significantly decreased by si-HELLS. Next, cell viability and proliferation were assessed by CCK8, colony formation, and EdU assays. As

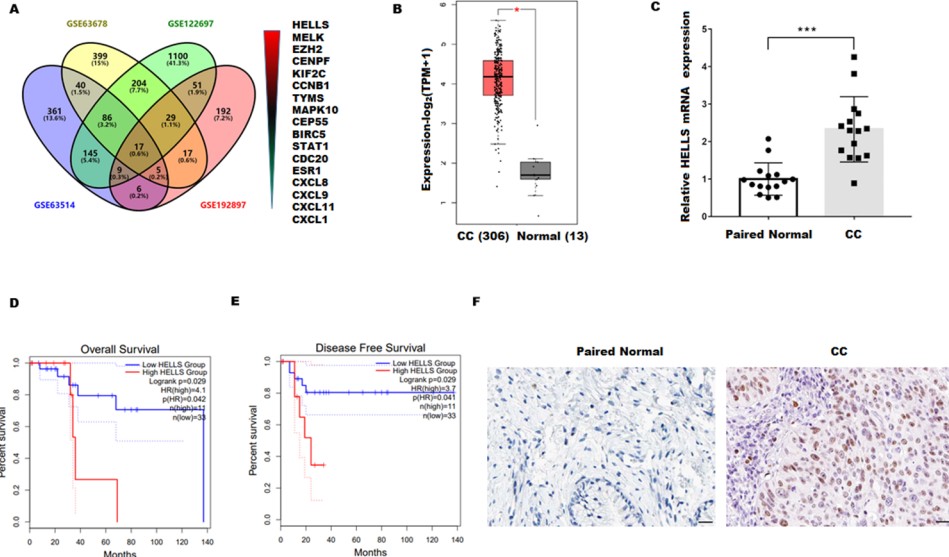

**Figure 1 HELLS was upregulated in cervicalcarcinoma.** (A) Venn analysis of DEGs of GSE63678, GSE122697, GSE63514, and GSE192897.(B) HELLS expression in cervicalcarcinoma from TCGA database. (C) HELLS expression in 15 cervicalcarcinoma tissues and paired normal tissues was analyzed by qRT-PCR. (D and E) Relationship between HELLS and OS (D), DFS (E). (F) HELLS expression in cervical carcinoma tissue and paired control was detected by IHC. Each data point was mean ± s.d. $*p < 0.05$, $***p < 0.001$.

expected, the knockdown of HELLS significantly reduced the cell viability (Fig. 3C) and proliferation ability (Figs. 3D and 3E) of SiHa cells.

## HELLS facilitates cervical cancer cell proliferation by inhibiting ferroptosis

To verify the function of HELLS in cervical cancer cell proliferation, the cell viability of HELLS knockdown SiHa cells was treated with or without the apoptosis inhibitor ZVAD-FMK or the ferroptosis inhibitor ferrostatin-1. Figure 4A showed that the decline in SiHa cell viability induced by HELLS knockdown is reversed by these two inhibitors. Moreover, we found that ferrostatin-1 was more efficient than ZVAD-FMK in restoring SiHa cell viability, which was decreased by HELLS knockdown (Fig. 4A). To further confirm the results of Fig. 4A, we performed EdU assay to measure the DNA synthesis and cell proliferation of SiHa cells with or without HELLS knockdown and inhibitors treatment. As shown in Fig. 4B, EdU incorporation was significantly reduced in SiHa cells transfected with HELLS siRNA compared with control siRNA. However, this reduction was partially reversed by ZVAD-FMK and ferrostatin-1, indicating that both apoptosis and ferroptosis were involved in the inhibition of SiHa cell proliferation by HELLS knockdown. Our data suggest that ferroptosis may play an important role in HELLS-mediated cell proliferation. To further confirm that HELLS promotes cell proliferation by inhibiting ferroptosis, HELLS-OE C4-1 cells were treated with or without the ferroptosis inducer erastin. Figure 4C showed that C4-1 cell viability was significantly decreased by erastin,

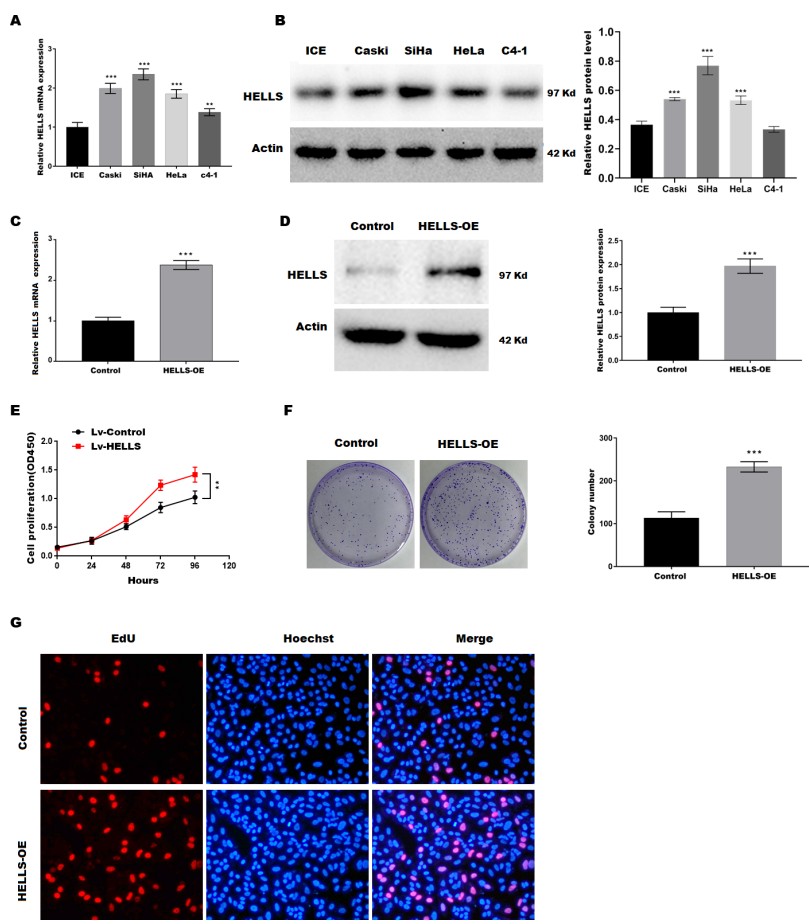

**Figure 2** **HELLS facilitates cervical cancer cell proliferation.** (A) HELLS in cervical cancer cell lines were analyzed by qRT-PCR. (B) HELLS in cervical cancer cell lines were analyzed by western-blotting. (C) The transfection efficiency of HELLS-OE was analyzed by qRT-PCR. (D) The transfection efficiency of HELLS-OE was analyzed by western-blotting. (E) C4-1 cells proliferation was assessed by CCK-8. (F) C4-1 cells proliferation was assessed by colony formation assay. (G) C4-1 cells proliferation was assessed by EdU assay. Each data point was mean ± s.d. **$p < 0.01$, ***$p < 0.001$. All experiments were repeated three times independently ($N = 3$).

while the effect was restored by HELLS-OE. This data was also confirmed by EdU assay (Fig. 4D). Taken together, these data suggest that HELLS facilitates cervical cancer cell proliferation by inhibiting ferroptosis.

## HELLS suppress cervical cancer cell ferroptosis by regulating Nrf2

Previous studies have confirmed that Nrf2, GPX4, and SLC7A11 are key regulators of ferroptosis. Therefore, we analyzed the correlation between the expression of HELLS and the expression of Nrf2, GPX4, and SLC7A11in cervical cancer using TIMER2.0 (http://timer.cistrome.org/). Figure 5A showed that HELLS expression was more significantly positively correlated with Nrf2, but not with GPX4 and SLC7A11, suggesting that HELLS may suppress cervical cancer cell ferroptosis by regulating Nrf2. To further confirm this conclusion, HELLS-OE C4-1 cells were treated with or without si-Nrf2.

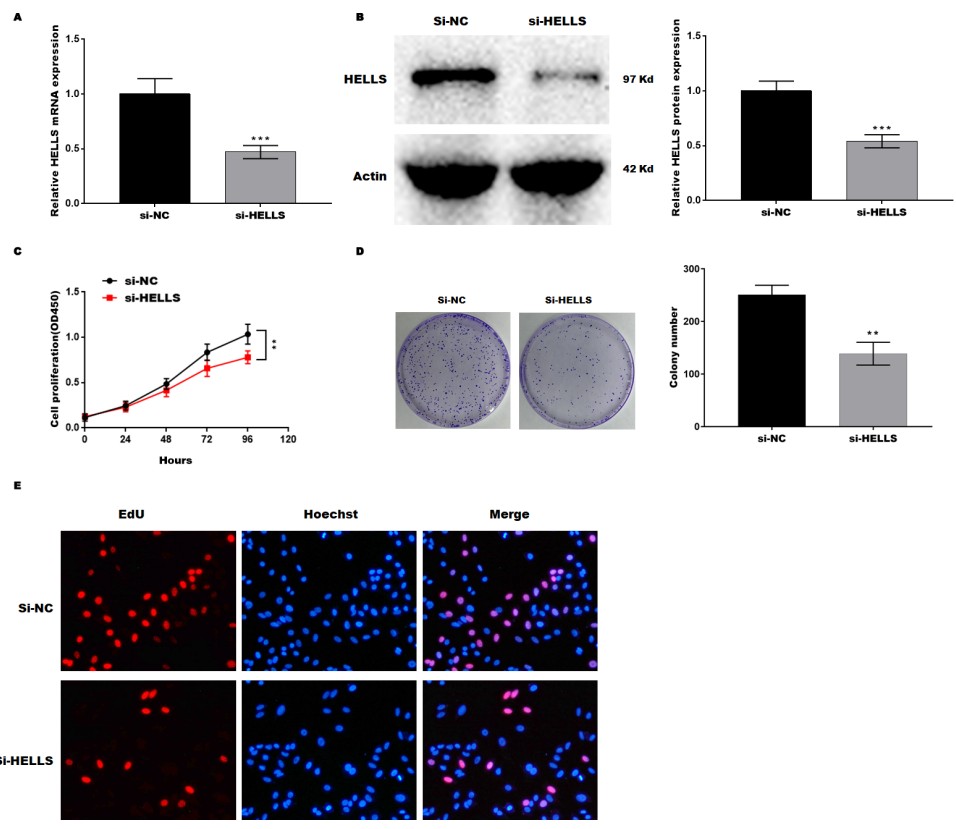

**Figure 3 HELLS inhibition promotes cervical cancer cell proliferation.** To investigate the effect of HELLS in SiHa cell, SiHa cells were transfected with si-HELLS. (A) The transfection efficiency of si-HELLS in SiHa cells was assessed by qRT-PCR. (B) The transfection efficiency of si-HELLS in SiHa cells was analyzed by western-blotting. (C) SiHa cells proliferation was assessed by CCK-8. (D) SiHa cells proliferation was assessed by colony formation assay. (E) SiHa cells proliferation was assessed by EdU assay. Error bars indicate the mean ± s.d. of three independent experiments. **$p < 0.01$, ***$p < 0.001$. All experiments were repeated three times independently ($N = 3$).

Figures 5B–5D showed that Nrf2 expression was significantly increased in HELLS-OE C4-1 cells and was markedly inhibited by si-Nrf2. Next, cell viability and cell proliferation were analyzed. Figure 5E showed that the promotion of cell viability by HELLS was significantly inhibited in the si-Nrf2 group. Similarly, the promotion of cell proliferation by HELLS was also significantly inhibited in the si-Nrf2 group (Figs. 5F and 5G). To further confirm the effect of Nrf2 on cell proliferation of HELLS-OE C4-1 cells, we performed EdU assay to measure the cell proliferation of HELLS-OE C4-1 cells with or without si-Nrf2 treatment. As shown in Fig. 5H, EdU incorporation was significantly increased in HELLS-OE C4-1 cells compared with control cells. However, this increase was partially reversed by si-Nrf2, indicating that Nrf2 was required for the promotion of cell proliferation by HELLS. These results suggest that HELLS enhances cervical cancer cell proliferation by regulating Nrf2 expression.

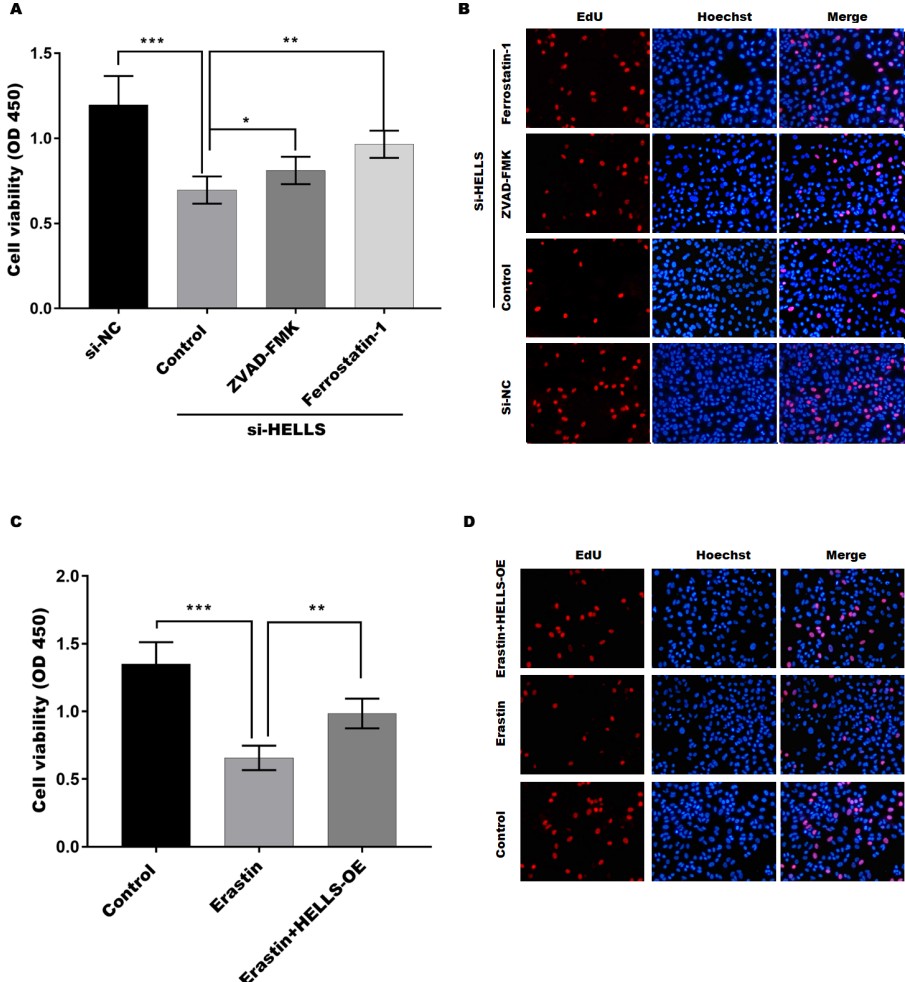

**Figure 4   HELLS facilitates cervical cancer cell proliferation by inhibiting ferroptosis.** (A) The cell viability of SiHa cells induced with si-HELLS in the presence or absence ZVAD-FMK (10 μM) or Ferrostatin-1 (0.5 μM) was analyzed by CCK-8 assay. (B) The cell proliferation of SiHa cells induced with si-HELLS in the presence or absence ZVAD-FMK (10 μM) or Ferrostatin-1 (0.5 μM) was analyzed by EdU assay. (C) The cell viability of C4-1 with HELLS-OE in the presence or absence erastin. (20 μM) was analyzed by CCK-8 assay. (D) The cell proliferation of C4-1 with HELLS-OE in the presence or absence erastin. (20 μM) was analyzed by EdU assay. Error bars indicate the mean ± s.d. of three independent experiments. $*p < 0.05$, $**p < 0.01$. $***p < 0.001$.

## DISCUSSION

Cervical carcinoma is one of the most prevalent tumor-related causes of death in women in the world. However, due to a lack of understanding of its mechanism, current treatment strategies are very limited, especially in the later stages of the patient. In the current study, we demonstrate that HELLS inhibits cervical cancer cells ferroptosis by promoting NRF2 expression, as evidenced by: (1) HELLS was upregulated in cervical carcinoma. (2) HELLS facilitates cervical cancer cell proliferation. (3) HELLS facilitates cervical cancer cell

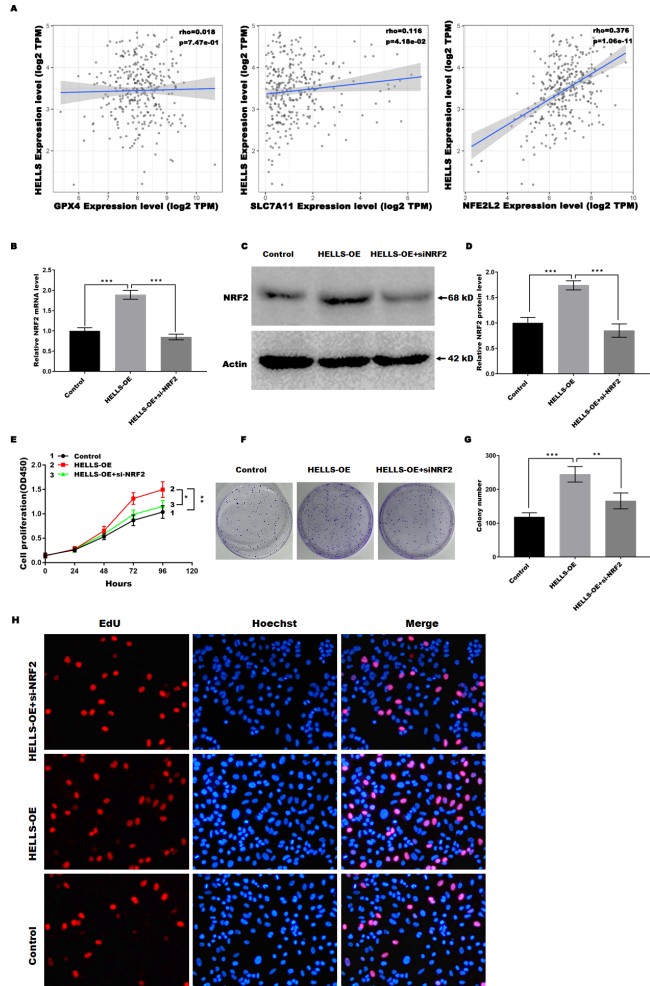

**Figure 5 HELLS suppresses cervical cancer cell ferroptosis by regulating Nrf2.** (A) Correlation between the expression of HELLS and the expression of Nrf2, GPX4, and SLC7A11 in cervical cancer was assessed by TIMER 2.0. (B) Nrf2 expression in C4-1 with HELLS-OE with or without si-Nrf2 was analyzed by qRT-PCR. (C and D) Nrf2 expression in C4-1 with HELLS-OE with or without si-Nrf2 was analyzed by western-blotting. (E) Cell viability of C4-1 with HELLS-OE with or without si-Nrf2 was analyzed CCK8 assay. (F and G) Cell viability of C4-1 with HELLS-OE with or without si-Nrf2 was analyzed by colony formation assay. (H) Cell viability of C4-1 with HELLS-OE with or without si-Nrf2 was analyzed by Edu assay. Error bars indicate the mean ± s.d. of three independent experiments. $*p < 0.05$, $**p < 0.01$. $***p < 0.001$. All experiments were repeated three times independently ($N = 3$).

proliferation by inhibiting ferroptosis. (4) HELLS suppress cervical cancer cell ferroptosis by regulating Nrf2.

Previous studies have revealed that HELLS, an enzyme involved in chromatin remodeling, is overexpressed in numerous types of cancer, including hepatocellular carcinoma, pancreatic cancer, and lung cancer (*Hou et al., 2021*; *Law et al., 2019*; *Yano et al., 2004*). *Law et al. (2019)* reported that HELLS inhibits multiple tumor suppressor genes including CREB3L3, XAF1, IGFBP3, FBP1, and E-cadherin in hepatocellular carcinoma through epigenetic silencing. *Hou et al. (2021)* reported that HELLS inhibition induces

tumor growth arrest and increases sensitivity to cisplatin by regulating TGFBR3. *Yano et al. (2004)* reported that the loss of function of HELLS may result in epigenetic deregulation and lead to the development of lung cancer. In accordance with these findings, our study also revealed an upregulation of HELLS in cervical carcinoma. However, the specific role of HELLS in the development of cervical carcinoma remains elusive. Numerous studies have demonstrated that HELLS enhances cancer cell proliferation and inhibits cancer cell death due to its primary function in DNA repair. *Yang et al. (2019)* reported that HELLS facilitates the proliferation of hepatocellular cancer cells by regulating the expression of CENPF. *Liu et al. (2019)* reported that HELLS was up-regulated in colorectal cancer and HELLS knockdown results in the inhibition of cell proliferation and colony formation. Our study further confirmed that HELLS promotes cervical cancer cell proliferation, aligning with the findings of *He & Liu (2022)*. However, the mechanism by which HELLS promotes the proliferation of cervical cancer cells is unknown.

Ferroptosis is a novel form of cell death characterized by lipid peroxidation and iron accumulation. In recent years, growing studies have confirmed that ferroptosis is closely related to cancer proliferation (*Lee et al., 2020*; *Ouyang et al., 2022*; *Wei et al., 2021*), including cervical cancer (*Wang et al., 2021a*; *Wu et al., 2021*). In line with these studies, our study also found that HELLS facilitates cervical cancer cell proliferation by inhibiting cell ferroptosis. Furthermore, our results revealed a significant positive correlation between the expression of HELLS and Nrf2. Additionally, our data revealed that the oncogenic role of HELLS in promoting tumor proliferation was attenuated upon Nrf2 inhibition, suggesting that HELLS suppressed ferroptosis of cervical cancer cells by regulating Nrf2 expression. This aligns with earlier research that has established the role of Nrf2 in the proliferation of various cancers through its regulation of ferroptosis (*Roh et al., 2017*; *Shin et al., 2018*).

The present study, while informative, exhibits certain limitations that warrant consideration in future research endeavors. Firstly, the absence of *in vivo* validation in animal models hampers the extrapolation of our findings to the clinical context. Expanding the investigation to incorporate robust animal models would bridge the gap between *in vitro* and *in vivo* observations. Secondly, the relatively limited size of clinical samples, both from the TCGA database and the 15 clinical specimens, underscores the need for larger cohorts to enhance statistical power and data robustness. Additionally, Nrf2, known for its robust antioxidant and anti-inflammatory capabilities, plays a crucial role in cancer development, given that inflammation is an important inducer for cancer. Studies have shown that chronic inflammation account for approximately 25% of cancer-causing factors (*Nigam et al., 2023*), However, this study did not explore the relationship between HELLS, NRF2, and inflammation. Despite the aforementioned limitations, this study highlights the potential role of HELLS, exploring its therapeutic implications and the development of HELLS inhibitors as novel treatment strategies is an avenue worth pursuing. Investigating the clinical relevance of HELLS as a diagnostic and therapeutic biomarker in cervical cancer patients should also be considered. Future studies should delve deeper into elucidating the molecular intricacies involved. Future investigations should prioritize *in vivo* validation, expand sample sizes, delve deeper into molecular intricacies, explore HELLS as a therapeutic

target, and assess its clinical relevance as a diagnostic or prognostic biomarker. Addressing these limitations will not only enhance our understanding of HELLS in cervical cancer but also have potential clinical implications for diagnostics and therapeutics.

In conclusion, in the current study, we demonstrate that HELLS inhibits Nrf2-mediated ferroptosis in cervical cancer cells, and our study provides a novel insight into the development of cervical cancer cells.

### Funding
The authors received no funding for this work.

### Competing Interests
The authors declare there are no competing interests.

### Author Contributions
- Weiwei Tie conceived and designed the experiments, performed the experiments, prepared figures and/or tables, and approved the final draft.
- Fenfen Ge analyzed the data, authored or reviewed drafts of the article, and approved the final draft.

### Human Ethics
The following information was supplied relating to ethical approvals (i.e., approving body and any reference numbers):

Research Ethic Committee of the Ningbo Medical Center Lihuili Hospital (DYLL2018079)

### Data Availability
The raw data are available in the Supplementary Files.

### Supplemental Information
Supplemental information for this article can be found online at http://dx.doi.org/10.7717/peerj.16451#supplemental-information.

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
