# Peer review of "Lymphoid-specific helicase inhibits cervical cancer cells ferroptosis by promoting Nrf2 expression"

_PeerJ, doi:10.7717/peerj.16451_

## Round 0.1 · original submission · Major Revisions

Dear authors, please refer to the reviewers' comments for further details. Additionally, know that Copyediting is not provided as a standard publication service. Please ensure the language in this submission is clear and unambiguous, grammatically correct and conforms to professional standards of courtesy and expression. Also, verify and confirm your statistical analysis and respective data.

**Language Note:** The Academic Editor has identified that the English language must be improved. PeerJ can provide language editing services - please contact us at copyediting@peerj.com for pricing (be sure to provide your manuscript number and title). Alternatively, you should make your own arrangements to improve the language quality and provide details in your response letter. – PeerJ Staff

Reviewer 1 ·

Basic reporting

the manuscript is very interesting but it presents some flaws that must be resolved. In particular:

Introduction: since NRF2 plays a key role in this manuscript, the multifaceted action of this transcription factor deserves to be highlighted. In fact, it is involved in several cancerous and non-cancerous diseases (see PMID: 36641100 , 36930785 , 37296999, 36632321, 37296665). This is an important point to add since it can further highlight the interesting results obtained by the authors. HELLS must also be introduced

Line 74: there is a repetition " three ferroptosis regulator factors, three ferroptosis negative regulatory factors,"

the number of replicates (N) must be added in the legend of each figure

Figure 2B and 3B: quantification of western blotting analysis must be shown. Moreover, molecular weights must be reported

Acronyms must be written in full length when mentioned for the first time

An accurate revision of syntax is recommended

Experimental design

no comment

Validity of the findings

no comment

Reviewer 2 ·

Basic reporting

The authors evaluated the implications of lymphoid-specific helicase in the inhibition of cervical cancer cell ferroptosis by promoting Nrf2 expression.
The subject is interesting and current but certain deficiencies in form and substance need to be improved.


-the results section of the abstract should be supported by some statistical data to prove the significance of the study

-it is recommended that keywords should also be generally used in non-abbreviated form for accuracy

-abbreviations used directly in the abstract should be used in unabbreviated form if used only once (CCK8) or abbreviated at the first mention then only the abbreviated form (Nrf2) should be used; the same principle applies to the main text which should be treated separately, please review the whole manuscript;

-bibliographic indexes inserted in the text (x) are singular self-standing structures and should not be pasted to any word, but a free space should be left.

-the introduction should be more detailed about the implications of HPV in types of infections, as well as some epidemiological data related to cervical carcinoma in different areas to highlight the burden created by it. I suggest checking and referring to: PMID: 30650666

Since Nrf2 and GPX4 have been discussed, it is important to detail but briefly the correlations between inflammatory processes and the cancer process: I suggest checking and referring to: PMID: 37321055

-it is advisable to discuss more in the introduction part which is slightly too poorly presented in relation to the complexity of the topic ferroptosis mechanisms. I suggest checking and referring to: PMID: 35408533

-The aim of the paper should be improved from the perspective of describing the contribution to the field under analysis and the elements of scientific novelty presented, especially since it is not the first manuscript evaluating these aspects.

-the number of patients included in the study is too small for the statistical power of the study -the contributions made by the present study should be better highlighted

-the date of ethics committee approval is also needed

-L117- capital letter is needed in the first word

-L147 and L161 - web page type bibliographic references are required for the computer software used

-It is also important to evaluate aspects of the therapeutic management of cancerous processes as well as the means of approaching young versus postmenopausal patients according to the risk of HPV infection. I suggest checking and referring to: PMID: 29167771 and PMID: 36677808

-the final part of the discussion needs to be improved/defined in terms of limitations and how it could be solved by future research perspectives -what future clinical impact the results of the present evaluation may have

Experimental design

Ok

Validity of the findings

Ok

Additional comments

None

---

## Round 0.2 · accepted · Accept

Dear authors, many thanks for your contribution and efforts. I am happy to let you know that I am now recommending your work for publication.

Reviewer 1 ·

Basic reporting

Acceptable as it is

Experimental design

Acceptable as it is

Validity of the findings

Acceptable as it is